# Delta-Notch Signaling: The Long and the Short of a Neuron’s Influence on Progenitor Fates

**DOI:** 10.3390/jdb8020008

**Published:** 2020-03-26

**Authors:** Rachel Moore, Paula Alexandre

**Affiliations:** 1Centre for Developmental Neurobiology, King’s College London, London SE1 1UL, UK; 2Developmental Biology and Cancer, University College London Great Ormond Street Institute of Child Health, London WC1N 1EH, UK

**Keywords:** neuron, neurogenesis, neuronal apical detachment, asymmetric division, notch, delta, long and short range lateral inhibition

## Abstract

Maintenance of the neural progenitor pool during embryonic development is essential to promote growth of the central nervous system (CNS). The CNS is initially formed by tightly compacted proliferative neuroepithelial cells that later acquire radial glial characteristics and continue to divide at the ventricular (apical) and pial (basal) surface of the neuroepithelium to generate neurons. While neural progenitors such as neuroepithelial cells and apical radial glia form strong connections with their neighbours at the apical and basal surfaces of the neuroepithelium, neurons usually form the mantle layer at the basal surface. This review will discuss the existing evidence that supports a role for neurons, from early stages of differentiation, in promoting progenitor cell fates in the vertebrates CNS, maintaining tissue homeostasis and regulating spatiotemporal patterning of neuronal differentiation through Delta-Notch signalling.

## 1. Introduction

During the development of the central nervous system (CNS), neurons derive from neural progenitors and the Delta-Notch signaling pathway plays a major role in these cell fate decisions [1,2,3,4]. The traditional view is that the cell presenting the ligand Delta at the cell membrane activates the Notch receptor in the adjacent cells (Notch *trans*-activation), delaying them from differentiating. Newborn neurons express Delta and Mindbomb (a ubiquitin ligase and Notch signalling pathway modulator) and hence are believed to activate the Notch signaling pathway in the surrounding tissue and maintain their neighbouring cells in a proliferative state [5]. However, recent works challenge this simplistic view and suggest that Notch and Delta interactions can also occur at the cell membrane within the same cell to inhibit the Notch pathway (Notch *cis*-inhibition) [6], or in specialised endosomes to enhance Notch activation [7,8]. 

The development of live-imaging approaches in the vertebrate nervous system has contributed to major breakthroughs in the field of neurogenesis and neuronal differentiation. This approach, which was initially developed in retina explants [9] and mammalian brain tissue, allows the visualization of biological processes such as neural progenitor divisions, in vivo generation of neurons, neuronal migration and axonal growth (for example [10,11,12,13,14,15]). Using this approach in mammalian embryonic brain tissue was critical to show that neurons can derive from neural progenitors through both asymmetric and symmetric divisions occurring at apical and non-apical locations of the neuroepithelium [10,11,12,13,14,16,17,18,19,20,21,22]. The neural progenitors that divide at the apical surface of the neuroepithelium are called apical progenitors and those that divide away from the ventricle are variously called non-apical [23], basal, or intermediate progenitors [24].

More recently, development of long-term live-imaging in the nervous system of zebrafish and chick embryos has been critical to elucidate the cellular and molecular mechanisms that regulate symmetric and asymmetric modes of progenitor division and generation of neurons [7,25,26,27,28,29,30]. Recent studies highlight the influential role that the daughters of neural progenitors may play in neural progenitor pool maintenance and tissue patterning. This review discusses how recent findings obtained in the vertebrate CNS reveal that newborn and differentiating neurons regulate both progenitor renewal and neuronal patterning via Delta-Notch signalling.

## 2. Neurons Derived from Asymmetric Divisions Can Influence Sister Cell Fate through Notch Signaling Pathway Activation

### 2.1. Neurons Inherit the Apical Attachment during Asymmetric Divisions 

During asymmetrically fated divisions in chick and zebrafish, each daughter cell inherits either the apical or basal attachment, and this asymmetric inheritance correlates with the adoption of distinct daughter cell fates, neuron and progenitor respectively (Figure 1) [25,26]. The neuronal daughter cell retains the apical domain containing the apical polarity protein Pard3 (previously known as Par3), while the progenitor daughter transiently loses the Pard3 protein and apical contact. These live-imaging studies also showed that nascent neurons remain integrated in the apical junctional belt for several hours following division. Later, neurons detach from the apical surface and move to the mantle layer at the basal surface of the neuroepithelium.

The correlation between inheritance of the apical domain and neuronal fate was unexpected considering that the majority of neural progenitors (which will continue to divide) contain an apical attachment and these observations directly contradicted the view at the time that a daughter cell that loses apical contact moves to the mantle layer and becomes a neuron. 

Nonetheless, in the mammalian brain, there is also evidence that inheritance of the basal process and loss of apical attachment correlates with progenitor fate [19,31]. At early stages of embryonic development, the neural progenitors that lose their apical attachment during division (10% of divisions) are able restore it in some cases [31]. There is also evidence that differentiating neurons are initially attached to the apical surface [32] suggesting that in mammals, like in zebrafish and chick, daughter cells inheriting the apical and basal cellular compartments during asymmetric divisions correlate with neuronal and progenitor fates respectively.

### 2.2. Recently Born Neurons Derived from Apical Progenitor Divisions May Activate the Notch Pathway in Sister Cells

Although early work in dissociated cell culture systems and some in vivo studies had shown an association between Par3 function and progenitor fate [33], experimental reduction in apical polarity protein (aPKC and Pard3) function has subsequently been shown to lead to a significant decrease in neurogenic divisions [25] and an overactivation of the Notch pathway [27], supporting the potential role for apical proteins in neuronal cell fate decisions. Further studies in zebrafish showed that Mindbomb plays a role downstream Pard3 function [27,30]. Mindbomb function is essential for Delta endocytosis and its loss-of-function blocks Notch trans-activation and leads to an increase in neuronal differentiation at the expense of neural progenitor fates [34,35]. During asymmetric division in the zebrafish telencephalon and chick spinal cord, the daughter cell committed to becoming neuronal inherits Mindbomb [27,30], while its sister cell (that does not inherit Mindbomb) activates the Notch signaling pathway and follows a progenitor fate [26,27]. This raised the possibility that, during asymmetric divisions, the neuronal daughter, through the inheritance of Pard3 and Mindbomb, activates Notch signaling in the sister cell. In zebrafish, the reduction of Pard3 function leads to Mindbomb symmetric inheritance [26,27] and symmetric proliferative divisions [25]. The observation that pairs of sister cells with decreased Mindbomb or Delta function are unable to activate the Notch pathway when surrounded by wild type cells in zebrafish [7,27], further supports the hypothesis that Notch activation depends on signals specifically provided by cells from the same lineage. In the mammalian brain the Notch receptor is enriched at the basal surface of the dividing cells, although we do not know whether the Notch receptor is asymmetrically inherited [10]. However, Mindbomb-1 mutant clones are able to activate the Notch signaling pathway when surrounded by wild type cells [5], raising the possibilities that Notch pathway activation in the mammalian brain can result from interlineage cellular interactions as proposed by Yoon et al. [5], or from intralineage cellular interactions mediated by the Mindbomb-2 function. Other works suggest that asymmetric activation of the Notch pathway in asymmetric divisions can result from the asymmetric inheritance of Sara-expressing endosomes by the progenitor daughter. Sara endosomes in both vertebrate and non-vertebrate systems carry Delta, Mindbomb [7] and Notch receptor [8], and it has been suggested that the inheritance of these endosomes can enhance the cell-autonomous activation of Notch signaling pathway in the progenitor fated cell.

### 2.3. Non-Apical Asymmetric Divisions—Do Newborn Neurons Influence Sister Cells Fates?

Intermediate progenitors, also called basal or non-apical progenitors, divide away from the apical surface and do not form apical or basal attachments. These progenitors, which were initially thought to be exclusive to the mammalian telencephalon, have now been reported in other brain regions and organisms [23,36,37,38,39,40]. There is evidence that intermediate progenitors can divide symmetrically to produce two neurons or asymmetrically to generate two neurons of different subtypes [37] and/or a progenitor and a neuron [24]. In the zebrafish spinal cord, for example, V2a and V2b neurons derive from a single asymmetric division that depends on Notch function [4,37]. The V2a daughter expresses the Notch ligand Delta C and Notch loss-of-function leads to an increase in V2a neurons at the expense of V2b neurons. This suggests a potential mechanism by which V2a may influence its sister cell to adopt a V2b fate in a Notch-dependant manner. 

In mammals, the basal radial glia progenitor subtype is also capable of dividing asymmetrically to self-renew and to produce neurons. Basal radial glia progenitors seem to preferentially inherit the basal process during division and express Hes1 [15] whose expression likely depends on Notch signalling pathway activation [41]. However, the underlying mechanisms that regulate intermediate progenitor renewal remain largely unknown, and so far there is little evidence that basal radial glia cell progenitor fate is influenced by signals provided by sister neurons. 

These studies overall provide indirect evidence that differentiating daughters derived from asymmetric divisions occurring at the apical or basal surface of the neuroepithelium have the potential to influence sister cell fates by activating the Notch signalling pathway in its sibblings.

## 3. Differentiating Neurons can Influence the Fate of Surrounding Cells during Apical Detachment

### 3.1. During Apical Detachment, Differentiating Neurons Influence Surrounding Cells to Maintain Progenitor Fates and Tissue Integrity 

Newborn neurons, across vertebrates and in different regions of the nervous system, have been shown to transiently retain the cellular process that attaches them to the apical surface of the neural tube (Figure 2) [25,26,32,42,43,44]. During differentiation, neurons detach from the apical surface of the neuroepithelium without disrupting the apical surface and compromising neuroepithelial tissue integrity. This is potentially achieved by neurons reducing the area of the apical end-foot prior to delamination [6] and neurons detaching from the apical surface through abscission of the apical end-foot [44].

Differentiating neurons are initially connected to their neuroepithelial neighbours at their apical processes through adherens junctions that include *N*-cadherin [6,32] and Notch signalling pathway regulates the neuronal apical detachment [6]. Initially the activation of Notch signalling is required (or maintained) in differentiating neuron to reduce the size of its apical area [6]. Notch signalling is then inhibited cell-autonomously (through *cis*-inhibition), which increases the expression of neuronal differentiation markers (such as Deltas and neurogenins) and reduces the localisation of *N*-cadherin to the neuronal apical end-foot [6]. The differentiating neuron expressing Delta-like 1 (Dll1) promotes progenitor fates in adjacent tissue by activating the Notch signalling pathway revealed by the expression of the Notch reporter gene Hes5 [6]. In the cortex, maintenance of Notch signaling in neurons following apical detachment also appears to be required for correct neuronal migration [45,46].

Notch1 signalling is required for the development and maintenance of radial glial cells [46], while reduction of Notch activity was previously shown to disrupt neuroepithelium integrity and increase neurogenesis [47,48]. There is also evidence that *N*-cadherin-based adherens junctions are critical to maintain tissue integrity and ensure correct rates of proliferation and differentiation [49]. However, the study reported by Baek and colleagues [6] supports a new hypothesis in which disruption of the neuroepithelial integrity due to Notch pathway inhibition results from the loss of *N*-cadherin without the reduction of neuronal apical end-foot area. It remains unknown whether Notch inhibition and an enlarged apical area would interfere with neuronal apical abscission, which has previously been suggested to potentially cause disruption of neuroepithelial tissue integrity [44,50]. 

### 3.2. Cellular Protrusions Developed by Differentiating Neurons Influence Neuronal Patterning in the Adjacent Tissue

Although it conventionally occurs between immediate neighbours, in *Drosophila*, Notch-Delta signalling has been shown to operate over larger distances to pattern mechanosensory bristles [51,52,53,54,55,56]. We recently showed that this can occur in the vertebrate neural tube [54]. All recently born neurons in the zebrafish spinal cord extend two long protrusions along the basal surface of the spinal cord that span several neural progenitors (Figure 2). These basal protrusions express high levels of Delta protein and Notch reporter activation occurs in the cells within their reach, suggesting that basal protrusions regulate Delta-Notch signalling pathway activation over long distances [54]. Spinal neurons initially differentiate with a sparse, periodic pattern [37,57,58,59,60] and never differentiate close in space and time [54]. We provided evidence that basal protrusions developed by differentiating neurons may spatially and temporally regulate the pattern of neuronal differentiation through long-range Delta-Notch-mediated lateral inhibition. This was further confirmed by mathematical modelling that showed the positioning and timing of neuronal differentiation cannot be explained by Delta-Notch signalling occurring between immediate neighbours but can be explained by the basal protrusions delivery of long-range Delta-Notch-mediated lateral inhibition [54]. 

Live imaging shows recently born chick spinal neurons extending highly dynamic, transient protrusions during apical detachment [61]. Similar processes have been described in the mouse neocortex, where basal progenitors (which are neurogenically committed) project transient, dynamic filopodia-like protrusions that contact radial glia processes [62]. As both radial glia cells and basal progenitors are molecularly heterogeneous and can be divided into subpopulations based on their Notch signalling pattern (Hes1 and/or Hes5 expression, for example) and expression of Delta (Dll1 and/or Dll3, for example) respectively, it suggests that basal progenitors may have the potential to activate Notch signalling pathway in radial glia cells through filopodia like protrusions [62]. However, this is yet to be proved. 

### 3.3. Where Do Delta-Notch Interactions Occur?

The subcellular localisation of Delta-Notch signalling in the majority of the contexts remains poorly characterised at the cellular and subcellular level. Depending on the vertebrate system and the moment of the cell cycle, Delta-Notch interactions have been suggested to occur closer to the apical surface [32], at the cell body and at the basal surface of the neuroepithelium [54].

In the mammalian brain for example, Notch receptor and a Notch cleaving protein, Presenilin1, are found overall apically and there is evidence that the Notch intracellular domain is cleaved at the apical surface of neural progenitors to be later translocated to the nuclei [32,63]. Delta antibody is also internalised at the apical surface of neuroepithelial cells [32], supporting the hypothesis that Delta-Notch interactions may take place at the apical surface at the adherens junctions. However, in zebrafish spinal cord, Delta D can be found in differentiating neurons in aggregates at the cell body and in the long cellular protrusions they develop at the basal surface of the neuroepithelium (see description in Section 3.2). A mathematical model developed by Hadjivasiliou and Moore et al. [54] to describe the long distance influence of basal protrusions on the spatiotemporal patterning of neuronal differentiation shows that Delta-Notch signalling mediated by basal protrusions is significantly more important than soma-soma signalling, suggesting that in this context at least, Delta-Notch signalling occurs predominantly basally. However, it remains unknown where Delta-Notch interactions occur, whether these locations are conserved across species, or whether they operate at different phases of the cell cycle and neuronal differentiation. 

In the developing retinal neuroepithelium, there is evidence that both Notch receptor and Delta ligands set up opposing spatial gradients of expression [64,65] and Notch activation correlates with the size of apical area and cell fates [66]. However, these observations need to be explored in greater detail. The development of better tools to visualise the in vivo dynamics and formation of gradients of Notch and Delta proteins will greatly assist in this area.

## 4. Conclusions

These works suggest that, from the moment of division, neuronal daughter cells are initially primed to activate the Notch signalling pathway in their sister cells and later, during apical detachment, in the surrounding tissue. While the location and timing of Delta-Notch interactions are better defined in *Drosophila* systems [56,67,68,69,70,71,72], less is known about the location and mechanisms of Delta-Notch interactions during neurogenesis and neuronal differentiation in the vertebrate neuroepithelium [32,54,63,66]. Delta is observed in basal protrusions [54] and at the apical surface [32], while Notch receptor is cleaved at the apical surface of neuroepithelial cells [32]. Thus, Notch activation appears to occur at multiple steps and for different purposes. Indeed, decreased proliferation of hippocampal progenitor cells is observed in conditional Notch mutants in adult mice, suggesting that the importance of Notch signaling may continue into adulthood [73]. Together, these studies illustrate the importance of refining signalling to certain areas of the cell body, but also suggest adaptation between different regions of the CNS. Understanding how this signalling changes over time during the initial period of neuronal differentiation remains a key question. We suggest that future work focussing on the mechanisms and subcellular locations of Delta-Notch interactions and how these change over the lifetime of both the organism and of an individual cell will give us a clearer understanding of signalling dynamics and how they influence cell fates.

## Figures and Tables

**Figure 1 jdb-08-00008-f001:**
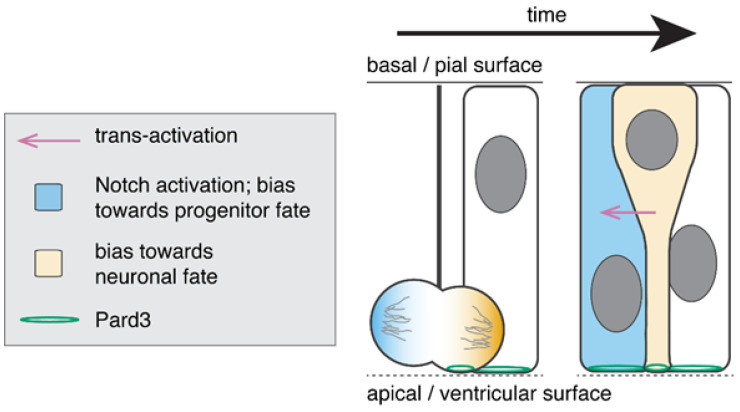
Recently born neurons may influence the fate of sister cells. Neural progenitors are polarised along the apico-basal axis of the neuroepithelium and localise apical polarity proteins such as Pard3 to the apical surface. Most neural progenitors divide at the apical surface of the neuroepithelium. A daughter that inherits the apical attachment (outlined by Pard3), inherits the Delta modulator Mindbomb and is likely to become a neuron. The daughter that transiently loses the apical attachment but retains the basal attachment is likely to remain a progenitor. Current evidence suggests that the neuronal daughter activates Notch signalling in its sister cell, promoting progenitor fate. However, exactly whether and when this occurs is not yet clear.

**Figure 2 jdb-08-00008-f002:**
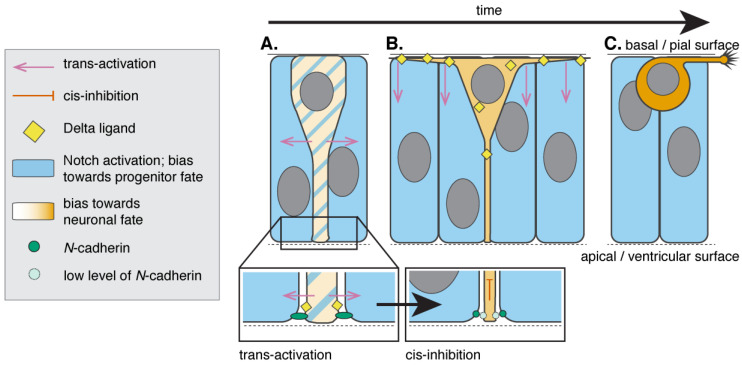
Delta-Notch signalling occurs at multiple steps during neuronal differentiation. (**A**) Recently born neurons (yellow) initially retain their attachment to the apical surface and are connected to their neighbours through adherens junctions that include *N*-cadherin (green circles). Prospective neurons require Notch activity (indicated by blue stripes) to reduce the size of the apical end-foot. There is evidence that at this point of differentiating neurons are capable of activating Notch signalling (blue) in the adjacent cells (Notch trans-activation). (**B**) Following reduction of the apical end-foot area, Notch signaling in the differentiating neuron is inhibited cell-autonomously, leading to reduction of *N*-cadherin localization at the apical end-foot (light green circle) and allowing apical process retraction. Meanwhile, differentiating neurons in the zebrafish spinal cord extend two long, transient processes along the basal surface of the neuroepithelium. Delta ligand (yellow diamonds) is enriched in the basal processes and Notch signalling (blue) is activated in the adjacent cells to prevent neuronal differentiation. Importantly, the basal processes can span several cell diameters and therefore contact cells that are not direct neighbours, activating Notch at a long distance. (**C**) Differentiating neurons finally retract the apical process and move to the basal surface of the neuroepithelium. The retraction of basal process and apical attachment precedes axon extension.

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
