# Peer review of "Delta-Notch Signaling: The Long and the Short of a Neuron’s Influence on Progenitor Fates"

_jdb, 2020, doi:10.3390/jdb8020008_

Round 1

Reviewer 1 Report

58-60: “During asymmetrically fated divisions each daughter cell inherits either the apical or basal attachment, and this asymmetric inheritance correlates with the adoption of distinct daughter cell fates (Figure 1; (Alexandre et al. 2010; Das and Storey 2012).”

These data were obtained in chick and zebrafish hindbrain and spinal cord. It should be mentioned and discussed that analyses in the mouse cortex by the lab of Fumio Matsuzaki (Shitamukai 2011; Fujita 2020)  suggests that more than 90% of divisions bisect the apical domain. This could be a region or species difference or may reflect a lack of temporal resolution in the mouse analysis if the apical reattachment is very fast.

64: “These live-imaging studies also showed that, after birth, neurons remain integrated in the apical junctional belt for several hours following division.”

“after birth” is misleading and could be replaced by “that prospective/nascent neurons remain integrated”

80 : “The observation that apical proteins promote neuronal fates is not supported by in vitro dissociated cell culture systems, where Par3 function has been associated with progenitor fates (Costa et al. 2008).”

Other in vivo evidence support  a role for Par3 as favoring the progenitor state, Costa et al., 2008 shows Par3 LOF in the cortex increasing differentiation and Afonso and Henrique, 2006 show in chick that overexpressed Par3 promotes the progenitor state.

82: “However, in the mammalian neuroepithelium, analysis of pairs of cells that result from oblique cleavages, the apical daughter expresses neurogenic markers Ngn2 /Tbr2 (Ochiai et al. 2009)”

That article does not analyze the cleavage plane but only the position of the daughter cell soma after mitosis. As mentioned before papers from the Matsuzaki lab suggest on the contrary that the vast majority of divisions in the mouse cortex do not bypass the apical plane. On the other hand, Ochiai et al. suggests that Neurog2/Tbr2 are more often (68-76%) expressed by the daughter cell soma localized more apically, but I am not sure how this observation serves the author’s argument.

84: “while the basal daughter has been reported to inherit the Notch receptor (Chenn and McConnell, 1995)” that study only shows a basal enrichment in dividing cells and not an asymmetric inheritance of Notch

86: “neural progenitors lose their apical attachment and then restore it following division at early stages of embryonic development (Fujita et al. 2020).”

This assertion is wrong, neural progenitors THAT lose their apical attachment restore it at early stages, but only 10% lose it in the wild type situation

87: “These studies provide indirect evidence that in mammals, like in zebrafish and chick, apical and basal daughter cells resulting from asymmetric divisions correlate with neuronal and progenitor fates respectively.”

This conclusion is much to strong and tends to merge the notion of apical foot inheritance and apical-basal position of the soma (relying on IKNM). Although there is some evidence in chick and zebrafish spinal cord that apical foot inheritance correlates with cell fate, whether this is necessarily connected with a stereotyped position of the soma shortly after division is not clear. On the hand, although the cleavage plane bisects the apical domain in most cases in the mouse cortex, it is possible that daughter cell nuclei migrate at different speeds towards the pial surface. These notions should be discussed.

95: “Further studies showed that Mindbomb plays a role downstream of asymmetric inheritance of Pard3 (Tozer et al. 2017; Dong et al. 2012)” Asymmetric inheritance of Par3 has not been shown in the chick, the Tozer et al study should not be referenced here and it should be mentioned that this was only shown in zebrafish.

103: “The observation that pairs of sister cells with decreased Mindbomb or Delta function are unable to activate the Notch pathway when surrounded by wild type cells (Dong et al. 2012; Kressmann et al. 2015), further supports the hypothesis that Notch activation depends on signals specifically provided by cells from the same lineage.” The authors should also mention work done in the mouse (Yoon et al., 2008) suggesting on the contrary that Mib1 deficient clones in a wild type tissue behave normally, suggesting extra-lineage Notch activation.

106: “Other works, however, suggest that an additional mechanism that leads to asymmetric activation of Notch pathway occurs during mitosis itself.”

Why “during mitosis itself”? As for Par3 and Mib1, asymmetric distribution in mitosis leads to asymmetric inheritance after division.

133: “These studies overall suggest that differentiating daughters derived from apical and basal asymmetric divisions have potential to activate the Notch signalling pathway in the sister cells.”

That conclusion is little vague. The whole paragraph is mostly descriptive apart from the lines about V2A/B neurons. The Youn et al. study should be mentioned.

More generally, I do not see how the introduction for part 2 which proposes a link between daughter cell position right after mitosis and cell fate, sheds any light on the interaction among sister cells developed in the following two sections

Titles for part 2 and 3 are essentially the same??

Author Response

Response to Reviewers

We would likelike to thank the reviewers for taking the time to review our manuscript and for providing such helpful comments. We are pleased that overall the feedback was positive and believe that we have addressed their recommendations in our resubmission.

In addition, we noticed that the first sentence in Section 2.1 (line 60) was ambiguous and so have reworded it to make it clear that the work discussed links the inheritance of basal attachment and loss of apical attachment to progenitor fate.

Reviewer 1:

Comments and Suggestions for Authors

58-60: “During asymmetrically fated divisions each daughter cell inherits either the apical or basal attachment, and this asymmetric inheritance correlates with the adoption of distinct daughter cell fates (Figure 1; (Alexandre et al. 2010; Das and Storey 2012).”

These data were obtained in chick and zebrafish hindbrain and spinal cord. It should be mentioned and discussed that analyses in the mouse cortex by the lab of Fumio Matsuzaki (Shitamukai 2011; Fujita 2020) suggests that more than 90% of divisions bisect the apical domain. This could be a region or species difference or may reflect a lack of temporal resolution in the mouse analysis if the apical reattachment is very fast.

We have reworded this sentence (line 60) to make it clear that the references provided specifically investigate chick and zebrafish.

Regarding the reviewer’s second comment:

We have no compelling evidence that there are substantial regional or species differences. First, the markers used to follow the inheritance of apical domain during progenitor divisions in chick/zebrafish (Par3-GFP) and mouse (ZO-1-GFP) localise to distinct apical domains. In addition, in the wild type zebrafish hindbrain, the majority of cell divisions occur with cleavages close to perpendicular and 20% of divisions generate daughters with asymmetric fates (neuron + progenitor), these values do not greatly differ from the values reported for mammalian brain development.

64: “These live-imaging studies also showed that, after birth, neurons remain integrated in the apical junctional belt for several hours following division.”

“after birth” is misleading and could be replaced by “that prospective/nascent neurons remain integrated”

We agree with the reviewer and have replaced “after birth” with “nascent” (line 67).

80 : “The observation that apical proteins promote neuronal fates is not supported by in vitro dissociated cell culture systems, where Par3 function has been associated with progenitor fates (Costa et al. 2008).”

Other in vivo evidence support  a role for Par3 as favoring the progenitor state, Costa et al., 2008 shows Par3 LOF in the cortex increasing differentiation and Afonso and Henrique, 2006 show in chick that overexpressed Par3 promotes the progenitor state.

We thank the reviewer for pointing this out, and this is now mentioned in the manuscript (lines 83-85).

82: “However, in the mammalian neuroepithelium, analysis of pairs of cells that result from oblique cleavages, the apical daughter expresses neurogenic markers Ngn2 /Tbr2 (Ochiai et al. 2009)”

That article does not analyze the cleavage plane but only the position of the daughter cell soma after mitosis. As mentioned before papers from the Matsuzaki lab suggest on the contrary that the vast majority of divisions in the mouse cortex do not bypass the apical plane. On the other hand, Ochiai et al. suggests that Neurog2/Tbr2 are more often (68-76%) expressed by the daughter cell soma localized more apically, but I am not sure how this observation serves the author’s argument.

We agree with reviewer’s comment and we have now removed this sentence.

84: “while the basal daughter has been reported to inherit the Notch receptor (Chenn and McConnell, 1995)” that study only shows a basal enrichment in dividing cells and not an asymmetric inheritance of Notch

We agree and have changed this (now line 87-89): “For example, the Notch receptor is enriched at the basal surface of the dividing cells, although we do not know whether the Notch receptor is asymmetrically inherited (Chenn and McConnell 1995).”

86: “neural progenitors lose their apical attachment and then restore it following division at early stages of embryonic development (Fujita et al. 2020).”

This assertion is wrong, neural progenitors THAT lose their apical attachment restore it at early stages, but only 10% lose it in the wild type situation

We clarified this point (line 89-91) “At early stages of embryonic development, the neural progenitors that lose their apical attachment during division (10% of divisions) are able restore it in the majority of the cases (Fujita et al. 2020)

87: “These studies provide indirect evidence that in mammals, like in zebrafish and chick, apical and basal daughter cells resulting from asymmetric divisions correlate with neuronal and progenitor fates respectively.”

This conclusion is much to strong and tends to merge the notion of apical foot inheritance and apical-basal position of the soma (relying on IKNM). Although there is some evidence in chick and zebrafish spinal cord that apical foot inheritance correlates with cell fate, whether this is necessarily connected with a stereotyped position of the soma shortly after division is not clear. On the hand, although the cleavage plane bisects the apical domain in most cases in the mouse cortex, it is possible that daughter cell nuclei migrate at different speeds towards the pial surface. These notions should be discussed.

We have reworked this section to clarify this point, the “apical and basal daughter cells” has now been replaced by “daughter cells inheriting the apical and basal cellular compartments during asymmetric divisions…” (lines 91-95).

95: “Further studies showed that Mindbomb plays a role downstream of asymmetric inheritance of Pard3 (Tozer et al. 2017; Dong et al. 2012)” Asymmetric inheritance of Par3 has not been shown in the chick, the Tozer et al study should not be referenced here and it should be mentioned that this was only shown in zebrafish.

We agree - we have removed this reference and specified that the work by Dong et al was performed in zebrafish (line 98-101).

103: “The observation that pairs of sister cells with decreased Mindbomb or Delta function are unable to activate the Notch pathway when surrounded by wild type cells (Dong et al. 2012; Kressmann et al. 2015), further supports the hypothesis that Notch activation depends on signals specifically provided by cells from the same lineage.” The authors should also mention work done in the mouse (Yoon et al., 2008) suggesting on the contrary that Mib1 deficient clones in a wild type tissue behave normally, suggesting extra-lineage Notch activation.

To address this concern we have now added the following sentence (line 112-115):

“In mammalian brain however, Mindbomb-1 mutant clones are able to activate the Notch signaling pathway when surrounded by wild type cells (Yoon et al, 2008), suggesting that Notch pathway activation can result from interlineage cellular interactions (Yoon et al, 2008) or through Mindbomb-2.”

106: “Other works, however, suggest that an additional mechanism that leads to asymmetric activation of Notch pathway occurs during mitosis itself.”

Why “during mitosis itself”? As for Par3 and Mib1, asymmetric distribution in mitosis leads to asymmetric inheritance after division.

We have changed this part of the text to:

“Other works suggest that asymmetric activation of the Notch pathway in asymmetric divisions can result from the asymmetric inheritance of Sara-expressing endosomes by the progenitor daughter. Sara endosomes in both vertebrate and non-vertebrate systems carry Delta, Mindbomb (Kressmann et al. 2015) and Notch receptor (Coumailleau et al. 2009) and it has been suggested that the inheritance of these endosomes can enhance the cell-autonomous activation of Notch signaling pathway in the progenitor fated cell.” (line 116-121)

133: “These studies overall suggest that differentiating daughters derived from apical and basal asymmetric divisions have potential to activate the Notch signalling pathway in the sister cells.”

That conclusion is little vague. The whole paragraph is mostly descriptive apart from the lines about V2A/B neurons. The Youn et al. study should be mentioned.

More generally, I do not see how the introduction for part 2 which proposes a link between daughter cell position right after mitosis and cell fate, sheds any light on the interaction among sister cells developed in the following two sections

Titles for part 2 and 3 are essentially the same??

We appreciate that our text was unclear and have updated the subheadings.

  1. Neurons derived from asymmetric divisions can influence sister cell fate through Notch signaling pathway activation (line 56)
  2. Differentiating neurons can influence the fate of surrounding cells during apical detachment (line 147)

Reviewer 2 Report

The submitted manuscript consists of a well structured focused review dealing with a scientific issue that requires elaborate experiments therefore increased amount of time in order to delineate the mechanisms through which neuronally differentiated progenitor cells determine the cell fate in sister cells. 

Overall the manuscript presents all relevant published data and it is well written. The only suggestion I could make in order to maybe provide an added value on the manuscript is the inclusion of a section describing the importance of Delta-Notch axis in neuronal-induced transdifferentiation in the context of diseases. This could add an important aspect of the processes that are described in the manuscript. 

Author Response

Response to Reviewers

We would likelike to thank the reviewers for taking the time to review our manuscript and for providing such helpful comments. We are pleased that overall the feedback was positive and believe that we have addressed their recommendations in our resubmission.

In addition, we noticed that the first sentence in Section 2.1 (line 60) was ambiguous and so have reworded it to make it clear that the work discussed links the inheritance of basal attachment and loss of apical attachment to progenitor fate.

Reviewer 2:

The submitted manuscript consists of a well-structured focused review dealing with a scientific issue that requires elaborate experiments therefore increased amount of time in order to delineate the mechanisms through which neuronally differentiated progenitor cells determine the cell fate in sister cells. 

Overall the manuscript presents all relevant published data and it is well written. The only suggestion I could make in order to maybe provide an added value on the manuscript is the inclusion of a section describing the importance of Delta-Notch axis in neuronal-induced transdifferentiation in the context of diseases. This could add an important aspect of the processes that are described in the manuscript. 

We appreciate the Reviewer 1 suggestion, and we agree that neuronal influence in neural progenitor fates can impact the disease development and therapies. However, there is not much evidence about this and we also think that discussing transdifferentiation will move us away from the scope of the present review.

Reviewer 3 Report

Comments

Manuscript Moore et al. summarizes current knowledge about the early stages of neurogenesis in embryonic development, in particular, the effect of neurons on the fate of neuronal precursors, according to the title. However, the topic considered in the article is narrower. The authors mainly consider the role of lateral inhibition of neurons through Notch-Delta pathways in the differentiation and organization of subpopulations of neuronal precursors. Undoubtedly, the Notch signaling pathway is recognized today as central to the spatial organization of neuronal precursors during embryonic development. In my opinion, this is either necessary to reflect in the title of the manuscript, or to consider broader the factors affecting the spatial organization of the precursors during prenatal neurogenesis.

Despite its importance, this pathway is not the only regulator the future spatial organization of the brain. For example, important regulators are BMP and Wnt signaling, Neurog2 etc. In addition, the authors don’t consider the important migration of neuronal precursors in the development of mammalian cortex, for example, into the hippocampus, where a neurogenic niche that is active in adulthood, as well as the relationship between embryonic and adult neurogenesis.

Minor concerns

The submitted manuscript is formatted sloppy including the use of different fonts and different design of references.

Author Response

Response to Reviewers

We would likelike to thank the reviewers for taking the time to review our manuscript and for providing such helpful comments. We are pleased that overall the feedback was positive and believe that we have addressed their recommendations in our resubmission.

Reviewer 3:

Manuscript Moore et al. summarizes current knowledge about the early stages of neurogenesis in embryonic development, in particular, the effect of neurons on the fate of neuronal precursors, according to the title. However, the topic considered in the article is narrower. The authors mainly consider the role of lateral inhibition of neurons through Notch-Delta pathways in the differentiation and organization of subpopulations of neuronal precursors. Undoubtedly, the Notch signaling pathway is recognized today as central to the spatial organization of neuronal precursors during embryonic development. In my opinion, this is either necessary to reflect in the title of the manuscript, or to consider broader the factors affecting the spatial organization of the precursors during prenatal neurogenesis.

Despite its importance, this pathway is not the only regulator the future spatial organization of the brain. For example, important regulators are BMP and Wnt signaling, Neurog2 etc.

We agree with the reviewer that our original title was quite broad. We have changed the title (line 2) to make it clear that the focus of this review is on Delta-Notch signalling, and we have also included a phrase in the abstract to this effect (line 18).

In addition, the authors don’t consider the important migration of neuronal precursors in the development of mammalian cortex, for example, into the hippocampus, where a neurogenic niche that is active in adulthood, as well as the relationship between embryonic and adult neurogenesis.

We have changed the text to mention neuronal migration (line 168), radial glial cell development and maintenance (line 170), and adult neurogenesis (line 262-264).

Minor concerns

The submitted manuscript is formatted sloppy including the use of different fonts and different design of references

We appreciate that this is frustrating for the reviewer, but unfortunately our text was altered when we uploaded the manuscript for review. We will work with the journal to ensure that this is corrected for the final version.

Round 2

Reviewer 1 Report

The authors have corrected the first version according to my concerns. However, sections 2.1-2.2 still need to be rewritten. The other sections are well written and easy to follow.

The construction of the 2 sections of 2.1 is very confusing, mixing molecular localizations and cellular events with no logic. It would be much easier to follow if the author fully described the cellular events before they move to potential molecular actors.

For instance , after 71 : « Subsequent studies have helped to elucidate the potential mechanisms downstream Pard3 asymmetric inheritance », we expect to read about these studies. Instead we read that Par3 is usually associated with a P fate (so the opposite) and then we move back to cellular observations (Shitamukai …), then back to a potential asymmetry of Notch, then back again to apical/basal endfeet… This section would gain in clarity if restricted only to cellular events.

In 2.2 : The Alexandre et al., 2010 and Dong et al., 2012 studies suggest that Pard3 is required for Mindbomb asymmetric localization which in turn ensures asymmetric Notch activation, which is not explained here (the data on Mib1 is developed after and not connected to the first affirmation). The data on Par3 in section 2.1 should be moved here and the interspecies differences could be discussed in light of the requirement of Par3 for Mib1 asymmetric localization (which has only been shown in zebrafish).

314 : “We suggest that future work focusing on the mechanisms and subcellular locations of Delta-Notch interactions and how these changes over the lifetime both of the organism and of an individual cell will give us a clearer understanding of signaling dynamics and how this influences cell fates.” There is a problem here with the syntax, “how these change”(s)? please rephrase.

Author Response

The authors have corrected the first version according to my concerns. However, sections 2.1-2.2 still need to be rewritten. The other sections are well written and easy to follow.

The construction of the 2 sections of 2.1 is very confusing, mixing molecular localizations and cellular events with no logic. It would be much easier to follow if the author fully described the cellular events before they move to potential molecular actors.

For instance, after 71 : « Subsequent studies have helped to elucidate the potential mechanisms downstream Pard3 asymmetric inheritance », we expect to read about these studies. Instead we read that Par3 is usually associated with a P fate (so the opposite) and then we move back to cellular observations (Shitamukai …), then back to a potential asymmetry of Notch, then back again to apical/basal endfeet… This section would gain in clarity if restricted only to cellular events.

We appreciate the reviewer suggestions and we have now clearly separated both sections.

In 2.2 : The Alexandre et al., 2010 and Dong et al., 2012 studies suggest that Pard3 is required for Mindbomb asymmetric localization which in turn ensures asymmetric Notch activation, which is not explained here (the data on Mib1 is developed after and not connected to the first affirmation). The data on Par3 in section 2.1 should be moved here and the interspecies differences could be discussed in light of the requirement of Par3 for Mib1 asymmetric localization (which has only been shown in zebrafish).

We have now addressed this concern between lines 128-144.

314 : “We suggest that future work focusing on the mechanisms and subcellular locations of Delta-Notch interactions and how these changes over the lifetime both of the organism and of an individual cell will give us a clearer understanding of signaling dynamics and how this influences cell fates.” There is a problem here with the syntax, “how these change”(s)? please rephrase.

This has been done.

Reviewer 3 Report

Authors successfully corrected flaws noted by reviewer

Author Response

Thank you Reviewer 3 for helping improve the manuscript.